# Neoadjuvant Treatment Strategies in Resectable Pancreatic Cancer

**DOI:** 10.3390/cancers13184724

**Published:** 2021-09-21

**Authors:** Aurélien Lambert, Lilian Schwarz, Michel Ducreux, Thierry Conroy

**Affiliations:** 1Medical Oncology Department, Institut de Cancérologie de Lorraine, CEDEX, 54519 Vandœuvre-lès-Nancy, France; t.conroy@nancy.unicancer.fr; 2Faculté de Médecine, Université de Lorraine, 54000 Nancy, France; 3Digestive Surgery Department, Rouen University Hospital, 1 rue de Germont, CEDEX, 76031 Rouen, France; lilian.Schwarz@chu-rouen.fr; 4Medical Oncology Department, Gustave Roussy, 39 rue Camille-Desmoulins, CEDEX, 94805 Villejuif, France; michel.ducreux@gustaveroussy.fr; 5Université Paris Saclay, FCS Campus, 91190 Saint-Aubin, France

**Keywords:** chemoradiotherapy, neoadjuvant chemotherapy, FOLFIRINOX, gemcitabine, nab-paclitaxel, pancreatic cancer, surgery

## Abstract

**Simple Summary:**

Only 10–20% of patients with newly diagnosed resectable pancreatic adenocarcinoma have potentially resectable disease. Upfront surgery is the gold standard, but it is rarely curative. After surgical extirpation of tumors, up to 80% of patients will develop cancer recurrence, and the initial relapse is metastatic in 50–70% of these patients. Adjuvant chemotherapy offers the best strategy to date to improve overall survival but faces real challenges; some patients will experience rapid disease progression within 3 months of surgery and patients who do not receive all planned cycles of chemotherapy have unfavourable oncological outcomes. The neoadjuvant approach is therefore logical but requires further investigation. This approach shows favourable trends regarding disease-free survival and overall survival but, in the absence of rigorous published phase III trials, is not validated to date. Here, we intend to provide a comprehensive analysis of the literature to provide direction for future studies.

**Abstract:**

Complete surgical resection is the cornerstone of curative therapy for resectable pancreatic adenocarcinoma. Upfront surgery is the gold standard, but it is rarely curative. Neoadjuvant treatment is a logical option, as it may overcome some of the limitations of adjuvant therapy and has already shown some encouraging results. The main concern regarding neoadjuvant therapy is the risk of disease progression during chemotherapy, meaning the opportunity to undergo the intended curative surgery is missed. We reviewed all recent literature in the following areas: major surveys, retrospective studies, meta-analyses, and randomized trials. We then selected the ongoing trials that we believe are of interest in this field and report here the results of a comprehensive review of the literature. Meta-analyses and randomized trials suggest that neoadjuvant treatment has a positive effect. However, no study to date can be considered practice changing. We considered design, endpoints, inclusion criteria and results of available randomized trials. Neoadjuvant treatment appears to be at least a feasible strategy for patients with resectable pancreatic cancer.

## 1. Introduction

Pancreatic cancer accounted for almost as many deaths (466,000) as new cases (496,000) worldwide in 2020 and is the seventh leading cause of cancer death in both sexes [1]. The prognosis of patients with pancreatic cancer has changed little over the past two decades: the overall 5 year survival rates in Europe and the USA remain dismal at 8–10% [2,3]. Pancreatic cancer is now classified into four categories: resectable tumors that have a high likelihood of achieving an R0 resection, borderline resectable tumors (BRPC) that are potentially resectable but have a high risk of an R1 resection with upfront resection, locally advanced unresectable pancreatic cancer (LAPC), and metastatic disease. This review on neoadjuvant strategies will concentrate on upfront resectable tumors.

Complete surgical resection is the cornerstone of curative therapy for resectable pancreatic adenocarcinoma; unfortunately, only 10–20% of newly diagnosed patients have potentially resectable disease. Upfront surgery is the gold standard, but it is rarely curative. Adjuvant chemotherapy has been shown to improve survival, and upfront surgery with subsequent adjuvant therapy is still the standard of care for all patients with resectable pancreatic cancer (RPC). However, about 10% of patients with RPC according to initial workup do not undergo resection because of occult metastatic disease found during surgical exploration [4]. Median survival with this multimodal therapy in patients with RPC was 17.7 months in a recent meta-analysis [5], and long-term survival remains limited, at approximately 10% with surgery and gemcitabine [6]. Recent randomized trials have reported an improved median disease-free survival (DFS) of 21.6–22.9 months with adjuvant mFOLFIRINOX (leucovorin, oxaliplatin, irinotecan, fluorouracil) or S-1 [7,8]. However, neither of these trials report how many patients were excluded from randomization due to chemotherapy contraindication or to poor performance status. After surgical extirpation of tumors, up to 80% of patients will develop cancer recurrence, and the initial relapse is metastatic in 50–70% of these patients. As a result of this, pancreatic cancer is often regarded as a systemic disease “ab initio” despite surgery [9,10], and there is growing interest in earlier use of systemic therapy.

## 2. Main Pitfalls of Adjuvant Therapy

Adjuvant treatment after macroscopically curative resection for pancreatic cancer is underutilized. Large retrospective series showed that 33–49% of patients who undergo curative resection do not receive adjuvant chemotherapy, mainly due to postoperative morbidity, comorbidity, prolonged recovery, patient refusal, and early disease recurrence [11,12,13,14]. In a US Surveillance Epidemiology and End Results series of 2440 patients from 2004 to 2013, only 7% of patients who underwent resection for pancreatic cancer completed their planned chemotherapy regimens, whereas 28% did not complete their courses and 65% received no adjuvant chemotherapy [15]. A similarly limited use of adjuvant chemotherapy has been reported in a recent population-based Dutch series. From 2008 to 2013, 46% of patients did not receive adjuvant chemotherapy following pancreatic resection. In the subgroup of patients >75 years, only 16% underwent adjuvant chemotherapy [16]. In the US National Cancer Database study from 2010 to 2014 of 18,470 patients with stage I or II resected pancreatic cancer, 27.5% did not receive any systemic therapy [17]. In another Dutch nationwide study, predictive factors for not receiving adjuvant chemotherapy were older age, worse performance status, postoperative complications, especially pancreatic fistula and postpancreatectomy hemorrhage, and lower annual surgical volume [18].

To our knowledge, incidence of early tumor recurrence before adjuvant chemotherapy has not been evaluated to date, but local recurrence may be detected at the first postoperative computed tomography (CT) scan or by elevated levels of the tumor marker CA 19-9 [19]. Postoperative CT or magnetic resonance imaging (MRI) scans and measurement of serum CA 19-9 levels may be useful for detecting early metastatic disease [7,20]. One retrospective study revealed that from a total of 957 patients with primary resection, 85 (8.9%) developed rapid disease progression within 3 months of surgery [14], suggesting that resection was not beneficial for these patients. Poor rates of completion of adjuvant chemotherapy is also a major problem, as many patients are unable to finish the planned cycles. The proportion of patients completing the full number of cycles has been shown to be 58–65% with gemcitabine [8,21,22,23] in all studies except PRODIGE 24 (in which it was 79%), 55% in the fluorouracil plus folinic acid group of ESPAC-3, 54% in the gemcitabine plus capecitabine group of ESPAC-4, 62% with S-1 (JASPAC-1), and 66% with mFOLFIRINOX [7,8,21,22,23]. This poor feasibility of completing adjuvant chemotherapy may be deleterious; completion of all cycles of planned adjuvant chemotherapy was identified as an independent prognostic factor in the ESPAC-3 trial [24].

## 3. Rationale for Neoadjuvant Therapy

Preoperative (neoadjuvant) treatment of localized pancreatic cancer is a logical strategy for a disease that is systemic at diagnosis in most patients. This multimodality approach may also overcome some of the limitations of adjuvant therapy. The improved response rates observed with FOLFIRINOX or gemcitabine with nab-paclitaxel and the recent positive results observed with neoadjuvant chemoradiotherapy (CRT) in BRPC [25,26] have encouraged their use in neoadjuvant settings. Neoadjuvant chemotherapy and/or CRT have theoretical and empirical advantages [27,28] and are given to patients with the following aims:-To eliminate presumed occult metastatic disease earlier after diagnosis and reduce distant relapse rates.-To give some patients time to allow for preoperative conditioning (nutrition, physical training, treatment of comorbidities and symptoms, etc.).-To increase complete resection (R0) rates-To evaluate the histological response to therapy.-To shrink tumors to smaller sizes and reduce involvement of vascular structures, and to facilitate R0 resection, which is associated with improved survival.-To downstage the tumor and reduce regional nodal disease and histological poor prognostic factors.-To reduce surgical complexity and postoperative complications.-To maximize the number of patients completing all cycles of chemotherapy and/or full doses of CRT.-To improve tolerance, resulting in a higher rate of treatment compliance and improved dose intensity.-To identify patients with rapidly progressive disease who are unlikely to benefit from resection and spare them from nonbeneficial surgery.-To test in vivo chemosensitivity and investigate novel sequential treatments and drug combinations.-Finally, to increase overall survival (OS) and quality of life of patients with RPC.

## 4. Main Concerns about Neoadjuvant Therapy

A surgery-first approach with adjuvant therapy may have benefits over neoadjuvant therapy. The main potential risk of neoadjuvant therapy is locoregional tumor growth or metastatic spreading of disease, and some patients may miss their “window” for curative-intent surgery. In metastatic disease settings, even with combination chemotherapy regimens, the risk of tumor progression is significant. Toxicities of neoadjuvant treatments may also result in decreased performance status, precluding surgery. Pancreatitis (e.g., after biliary stent placement) might be mistaken for tumor progression or unresectable disease. Some patients with inaccurate pretreatment staging, particularly those with missed peritoneal metastatic disease, may have inappropriate therapeutic sequences, especially those patients receiving neoadjuvant CRT. Increased postoperative complications and mortality rates may also be a significant risk. Potential overtreatment does not seem to be a significant problem, due to the small proportion of patients who are cured with standard treatments. So, the major point is: do we have level 1A evidence of evidence of the efficacy of neoadjuvant therapy through available literature and randomized trials?

Three main approaches have been developed:-Neoadjuvant chemotherapy, of which the main potential advantage is the earlier eradication of distant metastases that are already present at the time of the initial diagnosis. Reducing the delay between diagnosis and start of chemotherapy may reduce the metastatic rate and may improve the prognosis of patients.-Neoadjuvant CRT may also downstage the tumor, increase the R0 resection rate, and reduce the risk of local recurrences.-Combined neoadjuvant chemotherapy and preoperative CRT has also been tested, mainly in expert centers.

## 5. Meta-Analysis of Neoadjuvant Therapy

Numerous meta-analyses, a Markov decision analysis, and Bayesian network meta-analyses of nonrandomized prospective and retrospective studies have been performed. Most have mixed RPC, BRPC, and LAPC and have reported improved R0 rates, decreased incidence of lymph nodes metastases, and significant OS benefit with neoadjuvant therapy. However, their impact is limited due to their inherent selection biases, as nonrandomized studies report survival data of the subgroup of patients who underwent pancreatic resection and patients with disease progression are often excluded [29,30,31]. Moreover, retrospective studies are known to underreport toxicity outcomes. In a systematic review that only included studies of RPC, Bradley and Van der Meer reported six phase II studies on neoadjuvant therapy, pooling 371 patients [32]. The proportion of patients who had surgery was 76.08% (95% CI: 60.8–88.5). Overall, the analysis, based on eight other studies and 9197 participants, marginally favors neoadjuvant therapy across outcomes of R0 resection rates and 5-year survival rates. All available meta-analyses are literature-based and did not use individual patient-level data.

Two recent meta-analyses compared neoadjuvant treatment to upfront surgery followed by adjuvant chemotherapy. A meta-analysis from the Amsterdam Cancer Center [5] selected 38 studies (three randomized trials, nine phase I or II trials, 12 prospective cohort studies, and 14 retrospective cohort studies) including 3484 patients treated with radiation (29 studies) and/or chemotherapy, mainly with gemcitabine (26 studies), for BRPC or RPC. The overall resection rate was lower in patients who had neoadjuvant treatment than in those who had upfront surgery (66.0% vs. 81.3%; *p* < 0.001). It is important to note that 17.8% of patients who received neoadjuvant treatment did not subsequently undergo exploratory surgery; disease progression was the reason for not undertaking surgery in 64.4% of these patients. When reported by intention to treat, the R0 resection rate was not significantly higher with neoadjuvant treatment (58.0% vs. 54.9%; *p* = 0.088). For the 18 studies that reported the median OS of 857 patients with RPC, the median OS was 17.7 months with upfront surgery compared to 18.2 months with neoadjuvant treatment. In an intention-to-treat analysis on the whole population of patients with RPC and BRPC, the median OS was 18.8 months after neoadjuvant treatment versus 14.8 months after upfront surgery (significance level not reported). However, the authors discussed the possible limitations introduced by the inclusion of retrospective studies that may have included bias.

One important meta-analysis from the Ohio State Wexner Medical Center [33] selected only prospective randomized controlled trials; six trials including 850 patients were relevant. Of the six trials, four included patients with RPC, one included patients with BRPC, and one included patients with either RPC or BRPC. Two trials used gemcitabine-based neoadjuvant chemotherapy, while four used neoadjuvant CRT. There was no significant difference in overall resection rates among the two groups, but there was an increased R0 resection rate (risk ratio [RR] 1.51, 95% CI 1.18–1.93) and an increased pN0 rate (RR 2.07, 95% CI 1.47–2.91) in the neoadjuvant group. Moreover, a significant overall benefit was reported (pooled hazard ratio [HR] 0.73, 95% CI 0.61–0.86). In subset analyses, the pooled HR remained significant both in RPC (HR 0.73, 95% CI 0.59–0.91) and in BRPC settings (HR 0.51, 95% CI 0.28–0.93).

## 6. Conditioning and Monitoring of the Patient during Neoadjuvant Treatment

Neoadjuvant therapy first requires a histological or cytological diagnosis of pancreatic cancer, either via endoscopic ultrasonography guided fine needle aspiration/biopsy or by endoscopic retrograde cholangiopancreatography with intraductal biopsy/brush, and biliary stenting for obstructive jaundice. For pancreatic head cancer, biliary drainage should be reserved for patients with jaundice [34]. Before starting neoadjuvant therapy, plastic stents should be replaced by fully covered self-expandable metallic stents, since they can be exchanged or removed [35].

Careful preoperative nutritional evaluation and nutritional support are important to reduce chemotherapy-related toxicities and improve tolerance to chemotherapy. A retrospective study on 62 patients, mostly treated with neoadjuvant CRT, suggested that neoadjuvant therapy could aggravate nutritional status and hamper postoperative recovery [36]. Conversely, in another retrospective series of 199 patients with BRPC and mainly treated with neoadjuvant chemotherapy, preoperative high prognostic nutritional index was an independent prognostic factor for survival [37]. Therefore, maintaining good nutritional status during preoperative treatment is important to prevent patient weight loss and reduce malnutrition, which are associated with increased morbidity and mortality and reduced QoL [38].

Prior to neoadjuvant treatment, MRI with diffusion-weighted imaging and/or laparoscopy allows the detection and biopsy of small peritoneal and/or liver metastases [4,39] and may help to avoid unnecessary morbidity from a nontherapeutic laparotomy, especially in patients at increased risk of metastatic disease: those with body or tail tumors, markedly elevated CA 19-9 levels, large primary tumors, or large regional lymph nodes [35,40]. Tumor board meetings and visits with the surgeon and the anesthetist should be organized before neoadjuvant treatment, as they play an important role in multidisciplinary team cancer care of surgical patients.

Patients are usually required to have a performance status of ECOG of 0 or 1 to receive chemotherapy. Placement of a central venous access or peripherally inserted central catheter line is required, and where the FOLFIRINOX regimen is used, screening for dihydropyrimidine dehydrogenase deficiency, blood tests for measurement of total bilirubin and unconjugated bilirubin (or for UGT1A1 genotyping) and serum CA 19-9 levels, cardiology visits, and electrocardiograms are typical. The optimal combination of prophylactic antiemetics should also be delivered to optimize compliance with treatment.

Restaging of pancreatic cancer, including chest and abdominopelvic CT scans, should be repeated every two months and following neoadjuvant therapy to evaluate the response to treatment and to rule out both unresectability or distant metastatic disease. A RECIST partial response and a reduction in tumor volume are favorable prognostic factors [41]. However, all patients with stable disease and no extrahepatic progression should undergo surgical exploration. CA 19-9 levels should be evaluated every two months and compared with baseline levels, as a low posttreatment CA 19-9 level [41,42] is of prognostic value.

Treatment in a high-volume center (≥40 procedures annually) should be preferred, as compliance to international recommendations is better, operative mortality is lower, and OS is increased [18,43].

## 7. Major Surveys and Retrospective Studies

Among the large retrospective studies available, some are of particular interest. A recent analysis included 11,699 patients with stage II pancreatic cancer from the National Cancer Database between 2010 and 2017 who had undergone R1/R2 surgery [44]. This study shows a strong signal in favor of surgery, even for patients with an R1 status, compared with patients who received chemotherapy alone, particularly in the setting of neoadjuvant chemotherapy. However, those results should be considered in light of various factors, such as the imbalance between the treatment groups, the assessment of resectability, the criteria for which may vary from one center to another, and the simple fact that patients who receive chemotherapy alone generally have a poorer prognosis.

The R1 status of more than 1 mm, as defined by the Royal College of Pathologists [45] is still under discussion regarding its prognostic role, even in randomized controlled trials such as CONKO 001 [21], PRODIGE24 [7], or ESPAC-3 [46].

In another database analysis, 16,666 patients (14,012 upfront resection; 2654 neoadjuvant therapy) were retrieved (National Cancer Database from 2007 to 2015) [47]. It showed that patients with neoadjuvant therapy had a significantly (log-rank, *p* < 0.001) better median OS (27.9 months, 95% CI 26.2–29.1) and 5-year OS (24.1%, 95% CI 21.9–26.3%) than the upfront surgery group (median OS 21.2 months, 95% CI 20.7–21.6; 5-year survival 20.9%, 95% CI 20.1–21.7%). A similar retrospective cohort study of 19,031 patients from the US National Cancer Database compared neoadjuvant therapy to upfront surgery. Preoperative therapy was used in 1772 patients and downstaged 38% of cN1 patients to ypN0. It was also shown that half of the patients treated with upfront surgery initially considered node free at clinical staging (cN0) had node-positive tumors (pN1) on the final pathology report [48]. This indicates that downstaging with neoadjuvant therapy is undervalued. This also shows the need to question the preparation of resection specimens, especially examining resection margins by comparing R0 and R1 status in future trials, as this emerges as a likely discriminating factor in large retrospective studies [13,47]. The margins of the pancreas should be painted with a color code before blocks are taken. However, there is currently no standardization of the histopathological examination technique, thus inducing a bias. R0/R1 margins and lymph node status should be more carefully evaluated in future trials, using standardized methods.

Nevertheless, a study of 458 patients identified by the California Cancer Surveillance Program reported that neoadjuvant therapy is associated with a lower rate of lymph node positivity (45% vs. 65%; *p* = 0.011) and an improved OS (31 vs. 19 months; *p* = 0.018) [49]. A potential bias of this retrospective cohort is the absence of data on resectability rates and on dropout for patients considered for neoadjuvant therapy. Such results have also been confirmed by another study in a matched cohort of 191 patients, with a significantly longer median OS (23.1 months vs. 18.5 months, *p* = 0.043), a lower incidence of positive surgical margins (8% vs. 30%, *p* < 0.002), and less lymph node metastasis (45% vs. 78%, *p* < 0.001) found in patients receiving neoadjuvant therapy versus those undergoing upfront surgery [50].

Focusing on older adult patients, neoadjuvant therapy still has serious arguments in its favor, as reported by Rieser et al. [51]. In their single institution retrospective study, they showed a favorable trend for neoadjuvant chemotherapy (versus a surgery-first approach) among patients of 75 years old and over. Median OS was higher in patients treated with neoadjuvant chemotherapy than in those who received upfront surgery (24.6 vs. 17.6 months; *p* = 0.01). Moreover, patients with upfront surgery had a trend toward higher rates of major complications (38% vs. 24%; *p* = 0.06) compared with patients who received neoadjuvant therapy.

Other factors, such as the proportion of patients receiving postoperative chemotherapy ratio, rapid progressive disease, and tumor side, are also to be considered. As an example, a study of 15,237 patients showed that 33% of patients who underwent upfront resection did not receive postoperative systemic chemotherapy [13].

Focusing on the tumor location, Ocuin et al. reported from the US National Cancer Database that in 6523 patients undergoing distal pancreatectomy, those who received neoadjuvant chemotherapy had a longer median OS (28.8 vs. 22.0 months; *p* < 0.001). However, multiagent neoadjuvant chemotherapy regimens were associated with improved OS compared with multiagent adjuvant regimens (30.2 vs. 23.1 months; *p* < 0.001), whereas single-agent regimens were not associated with a survival benefit [52]. To date, there are no data suggesting that pancreatic tumor location should be a consideration in the decision of whether to give neoadjuvant chemotherapy.

The question of the postoperative complications rate has not yet been resolved. There are still contradictory trends, as presented in Table 1. A dedicated focus should be placed on this in future trials.

## 8. Randomized Trials of Chemoradiotherapy and Neoadjuvant Chemotherapy

After meta-analyses suggested that neoadjuvant treatment had an effect [33,56], a number of randomized trials were set up and have been completed and published at this time. The first was a small German trial comparing CRT using gemcitabine and cisplatin before surgery with upfront surgery in RPC [57]. The trial was stopped after inclusion of 73 patients; 66 patients were eligible for analysis. Radiotherapy was completed in all patients. Chemotherapy was changed in three patients due to toxicity. Tumor resection was performed in 23 patients in the surgery group and in 19 patients in the neoadjuvant group. The R0 resection rate was 48% in the surgery group and 52% in the neoadjuvant group (*p* = 0.81), and (y)pN0 was 30% versus 39% (*p* = 0.44), respectively. Postoperative complications were comparable between the groups. Median OS was 14.4 months in the surgery group versus 17.4 months in the neoadjuvant group (intention-to-treat analysis; *p* = 0.96). After tumor resection, median OS was 18.9 in the surgery group versus 25.0 months in the neoadjuvant group (*p* = 0.79). This worldwide first randomized trial for neoadjuvant CRT in pancreatic cancer showed that neoadjuvant CRT is safe with respect to toxicity, perioperative morbidity, and mortality. Nevertheless, the trial was terminated early due to slow recruiting, and the results were not significant. The second was an Italian trial, in which 93 patients were randomly allocated to treatment between 5 October 2010 and 30 May 2015 [58]. One center was found to be noncompliant with the protocol, and all five patients at this center were excluded from the study. Thus, 88 patients were included in the final study population: 26 in Group A, to receive surgery followed by adjuvant gemcitabine 1000 mg/m^2^ on days 1, 8, and 15 every 4 weeks for six cycles; 30 in Group B, to receive surgery followed by six cycles of adjuvant PEXG (cisplatin 30 mg/m^2^, epirubicin 30 mg/m^2^, and gemcitabine 800 mg/m^2^ on days 1 and 15 every 4 weeks and capecitabine 1250 mg/m^2^ on days 1–28); and 32 in Group C, to receive three cycles of PEXG before and three cycles after surgery. In the per-protocol population, six (23%, 95% CI 7–39) of 30 patients in Group A were event-free at 1 year, as were 15 (50%, 95% CI 32–68) of 30 in Group B, and 19 (66%, 95% CI 49–83) of 29 in Group C. The main grade 4 toxicity reported was neutropenia (two [11%] in Group A, four [19%] in Group B, none in Group C). Febrile neutropenia was observed in one patient (3%) before surgery in Group C. No treatment-related deaths were observed. Median OS was 20.4 months (95% CI 15.6–25.8) for patients in Group A, 26.4 months (95% CI 15.8–26.7) for patients in Group B, and 38.2 months (95% CI 27.3–49.1) for patients in Group C. In the per-protocol population of Group C, median OS was 39.8 months (95% CI 28.8–50.8). The trial was stopped at its phase II level because the standard of care in terms of multiagent chemotherapy changed during the inclusion period.

The third trial, PREOPANC-1, recently published by the Dutch Pancreatic Cancer Group, is much more solid in terms of evidence-based medicine [26]. Patients were treated either with surgery followed by six cycles of gemcitabine in the control group (127 patients) or with one cycle of gemcitabine, followed by CRT (radiotherapy: 36 Gy in 15 fractions + gemcitabine), followed by one cycle of gemcitabine and then surgery, followed by the last four cycles of gemcitabine (119 patients). Unfortunately, inclusion was not restricted to RPC but also included BRPC. Between April 2013 and July 2017, 246 eligible patients were randomly assigned; 119 were assigned to neoadjuvant CRT and 127 to upfront surgery. Median OS by intention to treat was 16.0 months with neoadjuvant CRT and 14.3 months with immediate surgery (HR 0.78; 95% CI, 0.58–1.05; *p* = 0.096), and the resection rate was 61% and 72% (*p* = 0.058), respectively. The R0 resection rate was 71% (51 of 72) in patients who received neoadjuvant CRT and 40% (37 of 92) in patients assigned to immediate surgery (*p* < 0.001). Neoadjuvant CRT was associated with significantly better DFS and locoregional failure-free interval as well as with significantly lower rates of pathologic lymph nodes, perineural invasion, and venous invasion. Survival analysis of the patients who underwent tumor resection and started adjuvant chemotherapy showed improved survival with neoadjuvant CRT (35.2 vs. 19.8 months; *p* = 0.029). The proportion of patients who suffered serious adverse events was 52% with neoadjuvant CRT versus 41% with immediate surgery (*p* = 0.096). Although the results of this large trial favor neoadjuvant treatment, the study was not considered to be practice changing outside of the Netherlands. The main reasons were the mixed population of resectable and borderline resectable patients, the use of gemcitabine monotherapy as an adjuvant chemotherapy, and lower OS rates than expected.

A Japanese trial evaluated the role of a combination regimen used only in Japan: gemcitabine and S-1 for two cycles followed by surgery. In both arms, patients received four cycles of adjuvant S-1 for 6 months after curative surgery, which is considered to be the standard of care after surgery for pancreatic cancer in Japan [59]. From January 2013 to January 2016, 364 patients were enrolled in 57 centers (182 to neoadjuvant treatment and 182 to upfront surgery). Of these, two were excluded because of ineligibility, therefore 182 patients in the neoadjuvant treatment group and 180 in the upfront surgery group constituted the intention-to-treat analysis set. The median OS was 36.7 months in the neoadjuvant treatment group and 26.6 months in the upfront surgery group; HR 0.72 (95% CI 0.55–0.94; *p* = 0.015). Grade 3 or 4 adverse events frequently (72.8%) observed in the neoadjuvant group were leukopenia or neutropenia. However, the resection rate, R0 resection rate, and operative morbidity were equivalent in the two groups. There was no perioperative mortality in either group. The authors concluded that this phase III study demonstrated significant survival benefits of neoadjuvant treatment in RPC. They added that the results indicated that neoadjuvant chemotherapy could be a standard of care for these patients [60]. However, the oral fluoropyrimidine S-1 is not available in Europe and the US. It is also difficult to understand why the patients in the neoadjuvant arm received two more cycles than the patients treated in the upfront surgery group. Finally, full publication of the data from this trial, currently available as an abstract from 2019 conference proceedings, is still awaited.

The most recent trial is a randomized trial that did not really evaluate the role of neoadjuvant treatment in pancreatic cancer, since it randomized 102 patients with RPC to different schedules of chemotherapy before surgery (the SWOG s1505 trial) [61]. Patients received one of the two active combination regimens used in the treatment of metastatic pancreatic cancer: modified FOLFIRINOX or gemcitabine/nab-paclitaxel. The primary outcome measure was 2-year OS, using a “pick the winner” design; for 100 eligible patients, accrual up to 150 patients was planned to account for cases deemed ineligible at central radiology review. From 2015 to 2018, 147 patients were enrolled and 102 were eligible; 55 in the FOLFIRINOX group (arm 1), 47 in the gemcitabine/nab-paclitaxel group (arm 2). For the FOLFIRINOX and gemcitabine/nab-paclitaxel groups, respectively, the 2-year OS was 41.6% and 48.8 and median OS was 22.4 months and 23.6 months. Neither arm’s 2-year OS estimate was statistically significantly higher than the a priori threshold of 40% (*p* = 0.42 in Arm 1 and *p* = 0.12 in Arm 2). Median DFS after resection was 10.9 months in Arm 1 and 14.2 months in Arm 2 (*p* = 0.87). The conclusion made by the authors is that it seems feasible and possible to obtain adequate safety and high resectability rates with neoadjuvant chemotherapy but that this trial brought little evidence that either regimen improves OS compared with the historical standard. In addition, the number of patients in this trial was limited and the trial used a randomized phase II design that is not completely adequate to evaluate the role of neoadjuvant treatment.

These results from randomized clinical trials (summarized in Table 2) suggest that although there is no definitive evidence for the efficacy of neoadjuvant therapy in RPC, this strategy should be evaluated in future trials.

## 9. Ongoing Randomized Trials

A number of phase II–III randomized trials are ongoing and are presented in Table 3. Some randomized trials using neoadjuvant regimens have been terminated early due to slow accrual. The NEOPAC phase III trial, comparing neoadjuvant gemcitabine and oxaliplatin to adjuvant gemcitabine, enrolled only two patients out of the 310 planned. The NEOPA trial had included 32 patients with BRPC or RPC in 4 years instead of the 410 patients initially planned [62]. The NEPAFOX study was closed early after enrollment of 40/126 patients [63]. Most of the ongoing studies use neoadjuvant gemcitabine-based CRT or FOLFIRINOX as neoadjuvant regimens versus upfront surgery, with OS as the primary endpoint. The timing of surgery after neoadjuvant chemotherapy—for example, whether to perform surgery after 4 or 8 cycles—still needs to be further investigated. If it is logical to think that if there is a benefit to neoadjuvant chemotherapy for tumors that can be operated on right away, this benefit should be possible for borderline tumors. To obtain reliable answers, it will be interesting to wait for the results of the PRODIGE 44/PANDAS study (NCT02676349) which seeks to evaluate the addition of radiochemotherapy after neoadjuvant treatment with mFOLFIRINOX in borderline pancreatic tumors.

## 10. Surgical Considerations Following Neoadjuvant Therapy for RPC

In the setting of neoadjuvant treatment, various points must be taken into consideration for surgical management: the rationale for staging laparoscopy and paraaortic lymph node sampling, the surgical approach and technique, and the importance of surgical margins, but also the possible impact of neoadjuvant therapy on the postoperative disease course. It should never be forgotten that reporting on neoadjuvant treatment studies is often done on intention-to-treat populations, whereas this is not always the case for adjuvant studies.

### 10.1. Staging Laparoscopy

The role of surgery in the management of RPC is not limited to surgical resection of the primary tumor. Indeed, different authors have shown the advantages of surgical exploration by laparoscopy. The main objective of preoperative workup is to track down occult metastases, to avoid futile resection and late introduction of systemic chemotherapy.

Despite remarkable technological advances in medical imaging modalities, approximately 20–50% of patients are found to have metastases at the time of surgery [69,70]. A new definition of radiologically occult pancreatic cancer metastases (ROMPC) has recently been proposed and corresponds to pancreatic cancer with metastases identified during surgery or within 6 months of resection [71]. Indeed, a nonnegligible rate of patients progress during neoadjuvant therapy or are unresectable during surgery for unknown metastasis, corresponding to the ROMPC situation. In the Dutch PREOPANC-1 trial [26], approximately 20% of patients were deemed ineligible based solely upon staging laparoscopy. Similar results were observed in two other recently published clinical trials (PREP-2/JSAP-05 trial, 28% [60]; Korean Phase III trial, 22% [25]). Based on these observations, some authors recommend the strategy of routinely incorporating baseline staging laparoscopy before neoadjuvant therapy for patients with PDAC [72]. The criteria usually used to indicate exploratory laparoscopy are tumor size >3 cm, body and tail tumor location, and CA 19-9 levels of >200 UI [73]. In the SLING Trial, Oba et al. demonstrated that staging laparoscopy with contrast-enhanced intraoperative ultrasound and indocyanine green fluorescence imaging was effective in such subgroups of patients, with the technique detecting ROMPC in 12 out of the 31 patients enrolled [74]. To help appropriate selection of patients for chemotherapy or surgery but also to decrease the perioperative morbidity of futile laparotomy, the routine use of staging laparoscopy needs to be discussed, even though it is considered still as an option by the National Comprehensive Cancer Network (NCCN) Guidelines [35].

### 10.2. Paraaortic Lymph Node Sampling—Is It Useful or Recommended Following Neoadjuvant Therapy?

The dismal prognosis of RPC following upfront surgery is, in part, related to the high frequency of distant lymph node metastasis, undiagnosed preoperatively. Lymph node metastasis of PDAC primarily involves peripancreatic nodes and eventually spreads to distant lymph nodes, including the paraaortic lymph nodes (PALN group 16b of the Japanese Pancreas Society Classification of Pancreatic Cancer) [75,76]. Although cross-sectional imaging has improved considerably during the last decade, the preoperative diagnosis of lymph node involvement in pancreatic cancer is still challenging and lymph node size remains the most frequently used criterion, although it has been shown to be not particularly sensitive for the detection of lymph node involvement [77]. To date, few data are available on neoadjuvant treatment and its impact on distant lymph node involvement or on oncological outcome. In the review by van Rijssen et al. [78], the number of patients with or without lymph node metastasis specifically receiving neoadjuvant therapy was not described. It was therefore not possible to analyze the impact of neoadjuvant treatment on outcomes in patients with distant lymph node metastasis. It could be hypothesized that neoadjuvant treatment may reduce the incidence of distant lymph node metastasis, especially paraaortic lymph node metastasis, but data are currently lacking.

### 10.3. Surgical Management

Usually, surgical resection is planned 4 to 8 weeks after the end of neoadjuvant treatment. Regarding surgical techniques, the main criteria for surgical quality are internationally accepted and should be respected, whether or not neoadjuvant treatment has been offered. In patients with adenocarcinoma located in the head of the pancreas, pancreaticoduodenectomy is the recommended surgical technique, and skeletonization of the superior mesenteric artery down to the adventitia on the anterior, lateral, and posterior borders is the standard of care [35,79]. The superior mesenteric artery–first approach seems to facilitate lymphadenectomy and optimize oncologic control of the retroperitoneal margin. The MAPLE-PD randomized trial (NCT03317886) aims to corroborate these results. In terms of lymph node dissection, it is well demonstrated that extensive lymphadenectomy does not improve long-term survival [80]. In patients with adenocarcinoma located in the left pancreas (body and/or tail), left splenopancreatectomy is recommended. Two techniques can be used, the so-called “standard” technique or a modified technique based on an antegrade approach (RAMPS: Radical antegrade modular pancreatosplenectomy), which aims to improve the retroperitoneal resection margin and lymphadenectomy [81]. The prospective randomized multicenter REMIND study aims to confirm these results (NCT03679169).

Regarding vascular contact/involvement, venous resection does not affect postoperative mortality although if it may slightly increase morbidity [82]. To optimize the chance of achieving an R0 resection, some surgical teams may even justify extreme positions such as performing routine venous resection during pancreatectomies, based on the results reported by Turrini et al. [83]. Indeed, in that study, patients who underwent portal vein or superior mesenteric venous venous resection but whose tumors did not infiltrate the vessel at final histology had significantly longer survival than patients in a matched control group who had pancreaticoduodenectomy without venous resection (42 months vs. 24 respectively, *p* = 0.02). Therefore, venous resection should not prevent surgeons from performing a pancreatectomy with curative intent, and patients with RPC should be managed by surgeons competent in vascular surgery, especially following neoadjuvant therapy.

### 10.4. Resectability and Resection Margins

The most recent analysis revealed that surgical exploration after neoadjuvant therapy was attempted in about 75% of patients but that among those patients only 72% were successfully resected (corresponding to 54% of all patients who received neoadjuvant therapy in intention-to-treat analysis) [84,85]. The goal of multimodal treatment is to achieve margin-free surgery. In surgical specimens, the most frequently involved margins are the posterior margin (retroperitoneal) (37%) and the portal vein–superior mesenteric vein margin (41%) [79]. The rationale for venous resection previously discussed is based on these observations. A meta-analysis by Schorn et al. showed that there is a lower risk of margins being classified as R1 after neoadjuvant therapy than after upfront surgery (RR 0.66, 95% CI 0.58–0.76, *p* < 0.00001) [86]. After neoadjuvant therapy for resectable PDAC, in resected patients the R0 rate was estimated to be around 70% [84] using the consensual guidelines of the College of American Pathologists [87]. As Verbeke et al. reported recently, pathologists should be aware that response after neoadjuvant treatment is frequently heterogeneous [88]. Indeed, the response to therapy within the tumor often results in multiple small foci of residual tumor cells in a background of mass-forming fibrosis and/or chronic pancreatitis, creating difficulties and inconsistencies in determining the appropriate tumor size or tumor-free margin measurement.

### 10.5. Postoperative Complications after Pancreatectomy Following Neoadjuvant Therapy

A retrospective study using ACS-NSQIP Targeted Pancreatectomy data identified 3748 operated patients, including 926 (25%) who had received neoadjuvant therapy (54.6% chemotherapy alone). The authors reported similar rates of postoperative mortality and overall complications between the two subgroups (neoadjuvant therapy versus upfront surgery), with a multivariable analysis revealing a reduced incidence of pancreatic fistula in those who had received neoadjuvant therapy (OR, 0.67; *p* < 0.001). The proposed mechanism for this effect of neoadjuvant therapy is through impairment of pancreatic function and induction of pancreatic fibrosis, thus making the pancreas more favorable for pancreatic ductal anastomosis [89]. The experience of the Verona group is consistent with the previous study, with patients receiving neoadjuvant therapy exhibiting a reduced incidence of postoperative fistula and hemorrhage versus patients who underwent upfront surgery. However, in the setting of the occurrence of such complications, the clinical burden was increased in patients who received neoadjuvant therapy [55]. There are currently no available prospective data about postoperative complications following neoadjuvant therapy with FOLFIRINOX, other than from the SWOG S1505 trial, in which no difference was observed between patients receiving FOLFIRINOX and those receiving gemcitabine/nab-paclitaxel, with no postoperative mortality observed in either group [84]. This reassuring approach is also confirmed in the study by van Dongen et al. where preoperative chemoradiotherapy did not increase the incidence of surgical complications or mortality [90].

### 10.6. Biological Borderline Pancreatic Cancer

According to the international consensus on definition and criteria of borderline resectable pancreatic ductal adenocarcinoma, we should embrace the yet unproven definition of a biological borderline pancreatic cancer with a CA 19-9 threshold of 500 units/mL or regional lymph nodes metastasis diagnosed by biopsy or PET-CT, which reflects the suspected distant metastasis [91]. The approach of Isaji et al. is a way to estimate distant locations not detected by the usual methods. Anticipating the presence of an advanced situation could allow us to better select patients and thus propose the best strategy. Perhaps neoadjuvant chemotherapy should be favored in patients with “biologically” advanced disease.

## 11. Conclusions

Resectable pancreatic cancer is a systemic disease and neoadjuvant chemotherapy is a logical approach with acceptable morbidity and no increased postoperative mortality. Numerous meta-analyses and large prospective and retrospective studies comparing neoadjuvant therapy with upfront surgery have reported tumor shrinkage, decreased node-positive disease, reduced rates of pancreatic fistula, and significantly improved overall survival rates in neoadjuvant therapy groups. However, the survival benefit with neoadjuvant therapy still needs to be investigated through high-quality published randomized studies. Moreover, benefit of adjuvant chemotherapy following neoadjuvant therapy and pancreatectomy is not well established and randomized data to answer this question are still lacking. All authors have read and agreed to the published version of the manuscript.

## Figures and Tables

**Table 1 cancers-13-04724-t001:** Early postoperative complications: neoadjuvant therapy versus upfront surgery.

Study	Difference in Postoperative Morbidity and Mortality	Result	Number of Patients	Trend
Mokdad et al. [13]	No	N/A	15,237	None
Ocuin et al. [52]	Yes	NAT patients: - Shorter length of stay (7.4 vs. 7.9 days; *p* = 0.011) - Lower rate of 30-day readmission (7.7% vs. 10.4%; *p* = 0.013) - Lower rate of 90-day postoperative mortality (3.0% vs. 5.6%; *p* = 0.005)	6523	Favors NAT
Rieser et al. [51]	Yes	US patients: - Higher rates of major postoperative complications (38% vs. 24%; *p* = 0.06) - Higher Comprehensive Complication Index totals (20.9 vs. 20; *p* = 0.03)	158	Favors NAT
Arrington et al. [44]	No	Patients who died within 2 months of diagnosis in the chemotherapy alone group or within 2 months of surgery in the surgical groups were excluded from the survival analyses	11,699	None
Cooper et al. [53]	Yes	NAT patients: - No significant difference in complications Neoadjuvant radiation associated with lower pancreatic fistula rates	1562	Favors NAT
Dhir et al. [54]	Yes	NAT patients: - Lower rate of 30-day readmission (univariable 5.5% vs. 7.4%, *p* = 0.006; multivariable OR 0.74, 95% CI 0.6–0.92, *p* = 0.006) - No significant difference in the length of stay and 30- or 90-day mortality - Higher rates of margin-negative resection (83% vs. 80%, *p* = 0.004) - No difference in OS (NAT vs. US: median 27 vs. 26 months, *p* = 0.02)	73,313	Favors NAT
Marchegiani et al. [55]	Yes	NAT patients: - Reduced incidence of pancreatic fistula and postpancreatectomy hemorrhage - Increased incidence of delayed gastric emptying and clinical burden	445	None

Abbreviations: N/A: not available; NAT: neoadjuvant therapy; OR: odds ratio; OS: overall survival; US: upfront surgery.

**Table 2 cancers-13-04724-t002:** Randomized trials of chemoradiotherapy and neoadjuvant chemotherapy.

Trial	Inclusion Criteria	Design of the Trial	Number of Patients	Primary Endpoint	Secondary Endpoints
Golcher et al. [57]	Resectable	Surgery (arm A) or CRT with gemcitabine + cisplatin followed by surgery (arm B)	33 33	Median OS: 14.4 17.4 (*p* = 0.96)	Time to progression: 8.7 (A) vs. 8.4 (B); *p* = 0.95 Tumor resection: 48% (A) vs. 52% (B); *p* = 0.81 (y)pN0: 30% (A) vs. 39% (B); *p* = 0.44
PACT-15 Reni et al. [58]	Resectable	Surgery followed by gemcitabine (arm A) or Surgery followed by 6 cycles of PEXG (arm B) or 3 cycles of PEXG before and after surgery (arm C)	26 30 32	Event free at 1 year: 6 (23%) 15 (50%) 19 (66%)	Median OS: 20.4 (A) (95% CI 14·6–25·8) vs. 26.4 (B) (15·8–26·7) vs. 38.2 (C) (27·3–49·1) (NS) 3-year OS: 35% (A) vs. 43% (B) vs. 55% (C) 5-year OS: 13% (A) vs. 24% (B) vs. 49% (C)
PREOPANC Versteijine et al. [26]	Resectable: Borderline resectable:	Surgery followed by 6 courses of gemcitabine (arm A) or 3 courses of gemcitabine, the second combined with 15 × 2.4 Gy radiotherapy, followed by surgery and 4 courses of adjuvant gemcitabine (arm B)	127 119	Median OS: 19.8 35.2 (*p* = 0.029)	R0 resection rate: 40% (A) vs. 71% (B); *p* < 0.001 Serious adverse events: 41% (A) vs. 52% (B); *p* = 0.096
PREOP-02/JSP05 trial Unno et al. [60]	Resectable	Surgery followed by 6 months of S-1 (arm A) or 2 cycles of gemcitabine + S-1 followed by surgery and 6 months of S-1 (arm B)	182 182	Median OS: 26.6 36.7 (*p* = 0.01)	Grade 3 or 4 adverse events (leukopenia, neutropenia): 72.8% (B) Resection rate, R0 rat and morbidity were similar in both arms
SWOG S1505 Sohal et al. [61]	Resectable	12 weeks of mFOLFIRINOX before and after surgery (arm A) Or 12 weeks of gemcitabine + nap-paclitaxel before and after surgery (arm B)	55 47	2-year OS: 41.6% 48.8%	Median OS: 22.4 (A) vs. 23.6 (B) Median DFS: 10.9 (A) vs. 14.2 (B); *p* = 0.87

Abbreviations: NS: not statistically significant; PEXG: PEXG (cisplatin, epirubicin, gemcitabine, capecitabine); Gem/gemcitabine; CRT: chemoradiotherapy; mFOLFIRINOX: leucovorin, oxaliplatin, irinotecan, fluorouracil; Median OS and DFS and Time to progression in months.

**Table 3 cancers-13-04724-t003:** Ongoing randomized clinical trials.

Trial	Inclusion	Modality and Regimens of Neoadjuvant Therapy	Planned Number of Patients	Primary Endpoint(s)	Secondary Endpoints
NEOPAC NCT01521702 [64]	R	Gemcitabine + oxaliplatin, 4 cycles or Upfront surgery	155 155	+15% in 1-year PFS	PFS; Histological response; OS; Complication rates after surgery; feasibility of adjuvant chemotherapy
NEOPA NCT01900327	R and BRPC	Gemcitabine + radiotherapy 50.4 Gy or Upfront surgery + adjuvant gemcitabine	205 205	+30% in 3-year OS	R0 resection rate; Frequency of toxicity events; Resectability rate; Rate of intraoperative irregularities; Postoperative complications; Disease progression during adjuvant therapy; DFS, QoL; First site of tumor recurrence
NorPACT-1 NCT02919787 [65]	R pancreatic head cancer	FOLFIRINOX, 4 cycles (+ 8 cycles of adjuvant chemotherapy) or Upfront surgery + mFOLFIRINOX (12)	54 36	Reduction in 1-year mortality from 25 to 5%	Overall mortality at one year; DFS; Histopathological response; Complication rate after surgery; Feasibility of chemotherapy; QoL; Health economics
PANACHE 01- PRODIGE 48 NCT02959879 [66]	R	FOLFIRINOX, 4 cycles (+ 8 cycles of adjuvant chemotherapy) or FOLFOX, 4 cycles (+ 8 cycles of adjuvant chemotherapy) or Upfront surgery (+ 12 cycles of adjuvant chemotherapy)	64 64 32	1-year survival Chemotherapy completion rate	Adverse events; Post-operative complications; Patients alive and without recurrence; R0 resection rate; QoL
PREOPANC-2 NL7094 [67]	R and BRPC	Chemoradiation 36 Gy/15 fractions with 3 cycles of gemcitabine, and adjuvant gemcitabine, 4 cycles or FOLFIRINOX, 4 to 8 cycles, and no adjuvant chemotherapy	184 184	OS	PFS; Locoregional progression-free interval; Distant metastases-free interval; Resection rate; R0 resection rate; Chemotherapy start rate; Chemotherapy completion rate; Toxicity; Post-operative complications; Radiologic response; Tumor marker response (CA 19–9 CEA); Pathologic response; QoL
PREOPANC-3 NCT04927780	R	FOLFIRINOX 8 cycles (+4 cycles in adjuvant setting) or Upfront surgery + FOLFIRINOX 12 cycles	189 189	OS	PFS; Distant metastases free survival; Locoregional PFS; Chemotherapy start date, Number of chemotherapy cycles and completion rate; Resection rate, R0 resection rate; N0 resection rate; pathological response; adverse events; CA 19-9 and CEA response: RECIST response; QoL
NEONAX NCT02047513 [68]	R	Nab-paclitaxel + gemcitabine 2 cycles (+ 4 cycles in adjuvant setting) or Upfront surgery + 6 cycles nab-paclitaxel + gemcitabine	83 83	DFS of ≥55% at 18 months in at least one arm	Safety; Morbidity and mortality; Toxicity; Resection rate; Tumor response; R0 resection rate, OS; Tumor recurrence; QoL
Alliance A021806 NCT04340141	R	mFOLFIRINOX 8 cycles, (+ 4 cycles in adjuvant setting) or Upfront surgery + mFOLFIRINOX (12 cycles)	176 176	OS	DFS; Time to locoregional or distant recurrence; R0 resection rate: pathological response; adverse events; QoL; Nutritional evaluation; Radiomics
PANDAS PRODIGE 44 NCT02676349	BRPC	Neoadjuvant mFOLFIRINOX regimen, with or without preoperative concomitant chemoradiotherapy (50.4 Gy + capecitabine)	45 45	histological R0 resection margin rate	Toxicites, proportions of resected patients, response rates to treatments, perioperative mortality and morbidity rates, OS, QoL, PFS

Abbreviations: BRPC: borderline resectable pancreatic cancer; DFS: disease-free survival; PFS: progression-free survival; R: resectable; mFOLFIRINOX: leucovorin, oxaliplatin, irinotecan, fluorouracil; QoL: Quality of Life.

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
