# Peer review of "Neoadjuvant Treatment Strategies in Resectable Pancreatic Cancer"

_cancers, 2021, doi:10.3390/cancers13184724_

Round 1

Reviewer 1 Report

The authors have addressed my comment in the revised version. I have no additional comment.

Author Response

Thank you for your help

Reviewer 2 Report

The paper has been improved and is suitable for publication

Author Response

Thank you for your help

This manuscript is a resubmission of an earlier submission. The following is a list of the peer review reports and author responses from that submission.

Round 1

Reviewer 1 Report

A Lambert et al prepared a review article entitled “Neoadjuvant treatment strategies in resectable pancreatic cancer”, describing and summarizing the current knowledge on neoadjuvant approaches in order to provide direction for future studies.

Overall, this is an outstanding review. The authors presented exhaustively the current clinical literature, including major surveys, retrospective studies, clinical trials, etc. but also many comprehensive sections about neoadjuvant therapy rationale and concerns, patient conditioning, which makes it far completer and more interesting than many reviews already published on the topic. There are no major flaws in the manuscript, and the manner in which it is laid out makes the paper highly readable and a very useful resource when considering a new neoadjuvant approach, which will likely be an often-referenced review article.

I have then no criticism on the content of the review and would simply recommend a careful double-checking of the potential spelling mistakes, typos, and repetitions (e.g. line 6 of page 1, line 2 of page 4).

Reviewer 2 Report

Dear collegues, 

many thanks for this extensive analysis of current knowledge about neo-adjuvant treatments in pancreatic carcinoma.

I agree that evidence is lacking despite the current mainstream philosophy is to propose neoadjuvant chemotherapy to most patients.

We all are aware that, despite evidence is more on pathology results than survival impact, future will be the "responder selection" to chemotherapy.

Anyway I don't understand the main focus of the paper. Is this a philosophic discussion about indication to neo-adj therapies or extensive statistical analysis?

I suppose that the paper is a discussion about potential evidences of the topic but the "discussion" itself, the most important part of the paper (due to the lack of a proper statistical analysis) is very poor.

Furthermore a revision of english is required to consider the paper for publication.

Reviewer 3 Report

The paper is of interest and gives a good overview on the topic of neoadj treatment for pancreatic cancer. 

The review is of interest. In my opinion, tables should be more informative on primary and secondary endpoints (not only OS). Please cut and simplify the result and design of the trial columns, make clearer columns creating primary and secondary endpoints (with correspondant results).
